# Precision Medicine on the Effects of Microbiota on Head–Neck Diseases and Biomarkers Diagnosis

**DOI:** 10.3390/jpm13060933

**Published:** 2023-05-31

**Authors:** Angelo Michele Inchingolo, Giuseppina Malcangi, Fabio Piras, Giulia Palmieri, Vito Settanni, Lilla Riccaldo, Roberta Morolla, Silvio Buongiorno, Elisabetta de Ruvo, Alessio Danilo Inchingolo, Antonio Mancini, Francesco Inchingolo, Gianna Dipalma, Stefania Benagiano, Gianluca Martino Tartaglia, Assunta Patano

**Affiliations:** 1Department of Interdisciplinary Medicine, University of Bari “Aldo Moro”, 70121 Bari, Italy; angeloinchingolo@gmail.com (A.M.I.); giuseppinamalcangi@libero.it (G.M.); dott.fabio.piras@gmail.com (F.P.); giuliapalmieri13@gmail.com (G.P.); v.settanni@libero.it (V.S.); l.riccaldo@studenti.uniba.it (L.R.); robertamorolla@gmail.com (R.M.); silvio.buongiorno@gmail.com (S.B.); studio.deruvo@libero.it (E.d.R.); ad.inchingolo@libero.it (A.D.I.); dr.antonio.mancini@gmail.com (A.M.); francesco.inchingolo@uniba.it (F.I.); studiobenagiano@gmail.com (S.B.); assuntapatano@gmail.com (A.P.); 2Department of Biomedical, Surgical and Dental Sciences, School of Dentistry, University of Milan, 20122 Milan, Italy; gianluca.tartaglia@unimi.it; 3UOC Maxillo-Facial Surgery and Dentistry, Fondazione IRCCS Ca Granda, Ospedale Maggiore Policlinico, 20122 Milan, Italy

**Keywords:** precision medicine, head and neck cancer, oral microbiota, biomarker, oral carcinoma, dentistry, oral disease, microbiome

## Abstract

Precision medicine using highly precise technologies and big data has produced personalised medicine with rapid and reliable diagnoses and targeted therapies. The most recent studies have directed precision medicine into the study of tumours. The application of precision medicine in the oral microbiota can be used both in the field of prevention and treatment in the strictly dental field. This article aims to evaluate the interaction between microbiota and oral cancer and the presence of biomarkers as risk predictors. Materials and Methods: A literature search of PubMed, Scopus, and Web of Science was performed analysing the various interactions between microorganisms, biomarkers, and oral cancer. Results: After screening processes, 21 articles were selected for qualitative analysis. Conclusion: The correlation between oral diseases/cancers and changes in the microbiota explains the increasing utility of precision medicine in enhancing diagnosis and adapting treatment on the individual components of the microbiota. Diagnosing and treating oral diseases and cancers through precision medicine gives, as well as economic advantages to the health care system, predictable and rapid management of the patient.

## 1. Introduction

In recent decades, great importance is being given to the study of the microbiota (MB) and microbiome (MM), which are considered responsible for complex human metabolism and disease onset [1].

MB refers to the complex of microorganisms (bacteria, viruses, protozoa, and fungi) that populate a specific environment at a specific period; MM, on the other hand, refers to the entire genetic pool contained in the MB that encodes 100 times more than the human genome [2,3].

The greatest concentration of microorganisms (more than one thousand species) is found in the gut.

The composition of the gut MB (GMB) is individual and varies according to genetics, geographic location, type and time of breastfeeding received, diet, and lifestyle (Figure 1).

Balanced (eubiosis) or altered (dysbiosis) conditions of the MB control the complex metabolic system and affect the immune system, neuroendocrine system, central nervous system, as well as behaviour (physical and psychic stress) (Figure 2) [4,5,6].

Precision medicine (PM) combined with the study of genomics (DNA), transcriptomics (RNA), epigenomics, proteomics, and metabolomics, using high-throughput technologies, has enabled the complex system of the MB to be studied in greater detail, with rapid, reliable, and less expensive methods [1]. 

Omics sciences prove to be fundamental in the study of the MB and in the identification of diagnostic, prognostic, and predictive biomarkers.

Present and future PM applied to the MB is mainly directed toward metabolomics and biomarker detection in addition to metagenomics, metatranscriptomics, and metaproteomics. Biomarkers studies are now considered fundamental in directing targeted and effective therapeutic plans for different diseases [7,8]. 

The correlation between alteration of the MB and the onset of disease is increasingly credited according to ”microbiome-based therapies” [9].

MB research and instrumental methodologies have not yet developed a unified and optimal protocol for analysis and application since the composition of the MB is different in every subject. Hence, there is a need to personalize treatments with reduced time and cost for diagnosis and treatment and the use of international databases [10,11].

Although several microbial systems (skin, vagina, oral cavity, upper and lower airways, etc.) are present in the human body, the most studied, because it is more complex, is the GMB, which seems to have an important role in homeostasis and state of well-being [4].

Two parameters are examined in the study of GMB microorganisms: α and β diversity [12,13].

Diversity α, calculated according to the Shannon index, Simpson index, and Chao index, examines the amounts and uniformity of microorganisms of different species present in a sample [3].

β-diversity, on the other hand, studies microbial communities from different environments and assesses their characteristics and differences [14,15,16].

The presence of a multiple and stable variety of microorganisms involved in diverse and complex metabolic functions characterizes a valid human health status [7,17]. 

More and more recent studies have correlated the GMB with the occurrence of cancer or disease in general. For example, colorectal carcinoma is characterized by the presence of increased *Fusobacterium nucleatum*, which determines its growth, prognosis (recurrence and metastasis) of the same, resistance to chemotherapy, and poorer survival [18,19,20,21].

Since 1994, the presence of *Helicobacter pylori*, a cause of chronic inflammation of the gastric mucous membranes, has been classified by the World Health Organization (WHO) and the International Agency for Research on Cancer as a class I carcinogen, associated with the occurrence of gastric cancer, the second leading cause of death worldwide, with 740,000 deaths per year [22,23,24].

In pre- and postmenopausal breast cancer, the composition of the GMB varies with different estrogenic production [25,26].

Poor diversity of microorganisms and the presence of severe dysbiosis were found in patients with leukaemia [27].

In addition to the study of the GMB, special attention is being given to the MB of the oral cavity, which is closely related to the occurrence of cancer diseases. Oral periopathogens such as *Fusobacterium nucleatum* and *Porphyromonas gingivalis*, as well as specific strains of *Aggregatibacter actinomycetemcomitans*, *Neisseria elongata*, and *Streptococcus mitis*, have been shown to play important roles in the onset of colorectal and pancreatic cancer [28,29,30]. 

*Capnocytophaga* and *Veillonella*, oral cavity bacteria, are numerous in lung cancer patients [31].

*Streptococcus* sp., *Peptostreptococcus* sp., *Prevotella* sp., *Fusobacterium* sp., *Porphyromonas gingivalis*, and *Capnocytophaga gingivalis* are microorganisms related to the occurrence of oral cancer, recognized among the most frequent malignancies [32].

The purpose of this scoping review was to be able to find the correlation between the human intestinal and OMB and the occurrence of cancer diseases. More and more scientific evidence has been found linking the possibility of chemotherapy choices and patients’ microbial variability in order to reduce the degree of toxicity and increase efficacy, resulting in the aim of finding a method of immunotherapy that can influence the variety of the GMB. Unfortunately, research on immunotherapy is limited and mainly based on mouse studies [33,34].

Further studies should be conducted on the composition of the MB and its influence on the efficacy, safety, and toxicity of chemo- and immunotherapy.

Artificial intelligence, together with “evidence-based” medicine, will enable improved diagnosis, prevention, screening, prognosis, and treatment plans (Figure 3) [11,35].

## 2. Materials and Methods

### 2.1. Protocol and Registration

This review was conducted according to the standards of the Preferred Reporting Items for Systematic Reviews and Meta-analysis (PRISMA) Extension for Scoping Reviews (PRISMA-ScR) [36].

### 2.2. Search Processing

We searched PubMed, Scopus, Web of Science, and ScienceDirect with a constraint on English-language papers from 1 January 2013 to 30 March 2023 that matched our topic. The following Boolean keywords were utilized in the search strategy: “precision medicine” AND (“microbiota” OR “microbiome”); (“oral microbiota” OR “oral microbiome”) AND “oral cancer treatment”; “biomarker” AND ”cancer” AND ”oral”. These terms were chosen because they best described the goal of our inquiry, which was to learn more about the prevention of head and neck cancers and potentially malignant lesions of the oral cavity using biomarkers and associated therapies.

### 2.3. Eligibility Criteria and Study Selection

We chose studies that looked at the effects of prebiotics, probiotics, and synbiotics on head and neck cancers and potentially malignant lesions as a form of prevention and treatment and MB modification in the course of cancer. The selection method was divided into two stages: (1) title and abstract evaluation and (2) full text examination. Any article that met the following criteria was considered: (a) human intervention studies (clinical trials); (b) supplementation with probiotics, prebiotics, symbiotic combinations; (c) studies assessing cancer biomarkers; (d) treatment was compared with a placebo, no intervention, or other interventions; (e) English language full text; (f) change in MB assessments were performed before and after the interventions using validated measures; and (g) changes MB assessments due to the presence of head–neck tumors and potentially malignant lesions. Publications that did not include original data (e.g., meta-analyses, research procedures, or conference abstracts, in vitro or animal studies) were excluded. The preliminary search’s titles and abstracts were retrieved and assessed for relevancy. For additional evaluation, full publications from relevant research were obtained. 

### 2.4. Data Processing

Author differences over the article selection were discussed and resolved.

### 2.5. Data Extraction

A standardized form was used to capture data on research design and locations, population characteristics (e.g., sex, age, presence of comorbidities), type of intervention and comparison, baseline measurements, and reported results. Each study was also evaluated for its handling of missing data and effect measurements. For extraction accuracy, two reviewers (F.P. and A.P.) worked separately; divergences were resolved by consensus. Because of the substantial variability in the treatments and outcomes reported, meta-analysis was not possible; consequently, papers were synthesized qualitatively.

### 2.6. Data Analysis

For homogeneous research, the fixed effect model was used, whereas for heterogeneous studies, the random effect model was used. In all analyses, the effect size was calculated using the standardized difference of means.

### 2.7. PICOS Criteria

Table 1 depicts the PICOS (Population, Intervention, Comparison, Outcome, Study design) criteria components, which include population, intervention, comparison, outcomes, and research design, as well as their use in this evaluation.

### 2.8. Study Evaluation

The article data were independently evaluated by the reviewers using a special electronic form designed according to the following categories: study type, aim of the study, materials and methods, and results.

## 3. Results

### Study Selection and Characteristic

The online database identified a total of 9155 studies (Scopus n = 2211, PubMed n = 3127, Web of Science n = 3817), and no articles were included through the hand search. After the deletion of 5484 duplicates, 3671 studies were screened by evaluating the title and abstract, focusing on the association between PM, MB, and head and neck cancer. There were 3538 articles which did not meet the inclusion criteria, leading to 133 records being selected. Subsequently, 0 non-retrieved records were excluded and then 112 reports were excluded because they did not meet the inclusion criteria. After eligibility, 21 records were selected for qualitative analysis. The selection process and the summary of the selected records were shown in Figure 4. The characteristics of the studies are summarized in Table 2.

## 4. Discussion

### 4.1. Cancer Diagnosis through Biomarkers

The latest studies have found that some of the microbial agents causing oropharyngeal cancer are the high-risk genotypes of the Human Papillomavirus (HPV): HPV 16, 18, and 33. Metagenomic shotgun sequencing (MSS) can overcome the limitations of existing commercial HPV detection assays. Ganly et al. employed MSS to assess HPV distribution in various body locations of human volunteers and discovered that many HPV types could not be identified using routinely used commercial kits and do not belong to the HPV types. Surprisingly, the HPV types discovered show significant organ tropism, and the oral HPV community varies from the vaginal HPV community [40]. These findings highlighted the potential that oral HPV strains that are undetectable to routine detection methods have a role in the genesis of human illnesses such as oral cavity squamous cell carcinoma (OC-SCC). In another study researchers found a connection between increased cigarette smoking and poor dental cleanliness. The oral MB was observed to be extensively changed in OC-SCC patients due to enrichment of the periodontal pathogens *Fusobacterium*, *Prevotella*, and *Alloprevotella* and the reduction in commensal *Streptococcus*. *Fusobacterium* and *Veillonella* were also more numerous in premalignant lesions (PML) than in controls [39]. Based on these marker genera, the oral MB may be split into two types: periodontal-pathogen-low and periodontal-pathogen-high. With better than 80% accuracy, this classification predicted PML and OSC-SCC. Aside from the three periodontal pathogens discovered in the samples, the cumulative abundance of all 14 periodontal pathogens discovered in the samples increased gradually throughout the sequence of negative controls PML-OC-SCC, with the pathogens being roughly three times as prevalent in OC-SCC as in negative controls. These data consistently indicate that periodontal infections are an independent risk factor in patients who do not have substantial OC-SCC risk factors [39]. Surrogate biomarkers are being used to assess the potential efficacy of chemoprevention. However, none of the markers have high levels of sensitivity, which may reduce specificity due to a large false-positive rate. The purpose of this study was to determine the efficacy of biomarkers and lifestyle dietary variables in chemoprevention [38]. The findings showed that p53 overexpression and expression in the para-basal layers of oral leucoplakia were inversely connected to the clinical response to betacarotene attained in this study. Daily consumption of green-yellow vegetables and fruits may also impact clinical reactions, albeit no significant differences were seen [38]. In this, study a digital, cell-based technique was used to analyze the immunohistochemistry hypoxia markers (hypoxia-inducible factor 1-alpha (HIF-1) and and pimonidazole (PIMO) in laryngeal squamous cell carcinoma (LSCC) and their influence on prognosis in patients with laryngeal cancer treated with accelerated radiation with or without carbogen breathing and nicotinamide (AR vs. ARCON) [37]. Findings suggest that it is possible to compute positive cell percentages and evaluate overlap of the two biomarkers in digitized sections of LSCC using open-source software. In another study, which examined tumor tissues from Iranian patients with OSCC, it was discovered that microRNAs miR-21-5p and miR-429 expressions were considerably greater in OSCC patients than in controls [42]. In order to untangle the interactions connected with the oral MB and its virulence factors, they employed 16S rDNA and metagenomic sequencing to assess the microbial composition and functional content in tumor tissue, non-tumor tissue, and saliva from 18 OSCC patients [41]. When compared with the other sample groups, the results show a larger number of bacteria from the phyla *Fusobacteria*, *Bacteroidetes*, and *Firmicutes* related to tumor tissue. Furthermore, saliva metaproteomics indicated a substantial rise in *Prevotella* in 5 OSCC patients, whereas *Corynebacterium* was predominantly related to 10 healthy patients [41]. Finally, they discovered adhesion and virulence factors linked to *Streptococcus gordonii*, as well as oral pathogens from the *Fusobacterium* genera, which were primarily detected in OSCC tissues. Based on these findings, we believe that the approaches used in this work may not only enhance OSCC diagnosis, but that the organisms and particular virulence factors found in tumor tissue may also be important markers for defining disease progression [41]. Regarding neoadjuvant immunotherapy, it is a novel therapeutic approach that has the potential to improve outcomes in patients with OCSCC. Immunotherapy generates varying degrees of toxicity, and a biomarker to predict response would be clinically useful in minimizing harmful effects in patients who are unlikely to benefit [43]. The purpose of this study is to link changes in FDG-PET/CT scans to primary tumor pathologic response and immunologic biomarkers in patients with OCSCC who have received neoadjuvant immunotherapy. In this study, FDG-PET/CT scans were performed before and after preoperative immunotherapy. FDG uptake changes during neoadjuvant immunotherapy were not linked to pathologic primary tumor response. In the majority of patients, freshly FDG-avid ipsilateral lymph nodes (LNs) were detected following neoadjuvant therapy but did not signal progressing illness or pathologically disease-positive nodes [43]. These findings imply that changing surgical plans in this patient is not advised, and that FDG-PET/CT has limited value as an early imaging biomarker for predicting pathologic response to preoperative immunotherapy for OCSCC [43]. 

HPV infection, smoking, and alcohol consumption are the three main conventional risk factors for developing HNSCC [58]; thus, Chan et al.’s study examined the relationship between OMB, HPV infection, conventional risk factors, and HNSCC [55]. Overall, 45.8% of HNSCC patients were negative for HPV infection, smoking, and alcohol consumption, suggesting dysbiosis of the MB as a potential etiological factor [55]. It is inferred that dysbiosis of the OMB is involved in the pathogenesis of OSCC [55]. *Fusobacterium* is involved in improving outcomes of patients with OSCC, especially in patients without traditional risk factors. Understanding how the OMM, HPV infection, and other risk factors for HNSCC interact will be crucial to understanding the pathophysiology of this disease [55].

There is mounting evidence that the OMB is critical in the development of oral cancer [59,60,61]. In the study by Li et al., three groups of samples from Chinese patients with oral cancer, patients with precancerous lesions, and healthy individuals were compared for their microbial composition using metagenomic sequencing [54]. While there was little variation in the three groups’ MB diversity, the OMB of patients with precancerous lesions was more diverse than that of oral cancer patients and healthy controls. Strain of Bacteroidetes was found in the phylum and was differently enriched in the samples of oral cancer [54]. The primary differentially enriched taxa at the genus level included *Prevotella*, *Peptostreptococcus*, *Carnobacterium*, and *Diastella*. *Prevotella intermedia* and *Peptostreptococcus stomatis* have distinct species levels of enrichment [54].

When compared with the healthy control group, the oral cancer group exhibits enhanced microorganism action in the manufacture of coenzyme A, phosphopantothenic acid, inosine 5′-phosphate degradation, and riboflavin. Increased production of dTDP-L-rhamnose, antigen 0, and unsaturated fatty acids are observed in the precancerous lesion group [54].

Overall, there were noticeable changes between the oral cancer group and the normal group in the oral bacterial profiles. Microbes can therefore be exploited as therapeutic targets and diagnostic markers for oral cancer [54].

Zuo et al. in 2020 also wanted to evaluate the association between oral microbes and head and neck cancer (HNC), as well as symptoms related to patients with HNC before surgical treatment [56]. Therefore, they recruited 56 patients with HNC and 64 control patients. Salivary samples were collected to determine microbial characteristics using 16S rRNA gene sequencing [56]. A decrease in health-related bacteria, such as *Peptococcus,* and a rise in potentially pathogenic bacteria, such as *Capnocytophaga* and other LPS-producing bacteria such as *Neisseria*, were seen in the oral microbiome of HNC patients [56]. Additionally, HNC-related symptoms in conjunction with salivary microorganisms such as *Capnocytophaga* may be employed as a noninvasive technique for screening, identification, and treatment monitoring of HNC [56].

Growing evidence suggests that the GMM influences the efficacy and toxicity of radiotherapy by modulating immune signaling. 

The study by Al-Qadami et al. aims to evaluate the associations between the pre-radiotherapy GMB and the severity of radiotherapy-induced oral mucositis (OM) and the risk of recurrence in HNC patients [57]. In the present study, the most common bacteria in the GMB were identified and divided according to the severity of OM. *Bacteroides* (40%), *Parabacteroides* (7.8%), *Faecalibacterium* (6.9%), unclassified *Ruminococcaceae* (6.8%) and unclassified *Clostridiales* (4.7%), whereas in the G3-4 OM group, *Bacteroides* (41. 9%), *Faecalibacterium* (7.9%), unclassified *Ruminococcaceae* (7.2%), *Prevotella* (5.5%) and unclassified *Lachnospiraceae* (4.2%) were identified. Therefore, *Eubacterium*, *Victivallis*, *Ruminococcus*, *Oxalobacter*, unclassified *Victivallaceae*, and unclassified *Desulfovibrionaceae* were significantly increased in patients with OM G3-4 [57]. The present study evaluated the GMB in relation to tumour recurrence. The most abundant genera among patients without recurrence were *Bacteroides* (39%), *Faecalibacterium* (8.9%), unclassified *Ruminococcaceae* (7.2%), *Parabacteroides* (5.9%), and *Prevotella* (4.9 percent) compared with *Bacteroides* (50%), unclassified *Clostridiales* (6.4%), unclassified *Ruminococcaceae* (6.0%), *Parabacteroides* (5.7%) and *Blautia* (4.5%) in the group of patients with relapse [57]. These microbes would appear to modulate the antitumor response to immunotherapy by enhancing the expansion and function of CD8+ T cells [57].

A pre-treatment MB enriched in *Eubacterium*, *Victivallis* and *Ruminococcus* is associated with severe OM. In contrast, higher relative abundance of immunomodulatory microbes *Faecalibacterium*, *Prevotella* and *Phascolarctobacterium* was associated with lower risk of tumour recurrence [57].

By controlling systemic inflammatory responses, the GMB’s makeup may have an impact on the pathogenesis of OM. Pro-inflammatory gut bacteria, for instance, could increase inflammatory processes in the oral cavity and lead to more severe OM if they were present in the GMB following treatment exposure. By contrast, intestinal homeostasis is promoted by gut bacteria with anti-inflammatory capabilities, and this reduces systemic inflammatory impulses, which results in moderate OM [57].

### 4.2. Microbiota and Head and Neck Cancer

In 2018, head and neck cancer (HNC) was the seventh most common cancer [32]. It is well known that cancer has a multifactorial aetiology [62,63]. Periodontal disease should also be regarded as a risk factor: in the 1940s, Cook et al. showed that periodontitis was a risk factor for the development of oral cancer, because of the chronic inflammation generated by microbes [64].

The complex human microenvironment includes the MB, which influences several physiological processes and the emergence of diseases through the interaction between microbes and hosts. The OMB significantly impacts the initiation and development of toumors. Increasing numbers of studies are showing a link between oral microbiota and oral cancer, particularly oral squamous cell carcinoma (OSCC) [15]. According to Li et al., oral bacteria found in OSCC are distinct from those in the tissues of people with health conditions. Particularly in patients with OSCC, there is a significant spread and abundance of *Fusobacterium* and a concomitant reduction in *p_Firmicutes* and *p_Actinobacteria* [53]. More generally, there has been a large rise in *p_Fusobacteriia*, *o_Fusobacteriales*, and *g_Fusobacterium*, as also supported by previous studies [65]. The most prominent distinctions may be found in *p_Actinobacteria*, *p_Firmicutes*, *c_Fusobacteriia*, *o_Fusobacteriales*, *f_Fusobacteriaceae*, and *g_Fusobacteriu*. Apart from *p_Actinobacteria*, this study found that five unique oral microorganisms have high confidence and may be used to predict clinical diagnosis and prognosis [53].

The occurrence of cancer by the action of the oral MB (OMB) is due to three basic mechanisms: chronic inflammation, cell proliferation, and the production of carcinogens [66,67,68,69].

The development of oral cancer appears to be a consequence of oxidative and nitrative stress produced by Reactive Oxygen Species (ROS)/Reactive Nitrogen Species (RNS) (nitrosamines, nitrates, NO3 and nitrites, NO2) free radicals. The higher concentration of ROS and RNS reduces the activity of salivary antioxidant systems, causing oxidative damage to proteins and DNA and increasing oncogenic properties [70,71].

Chronic inflammation produces various cytokines, such as matrix metalloproteinases MMP-8 and MMP-9, interleukin-1β (IL-1β), IL-6, IL-17, IL-23, and tumour necrosis factor-α (TNF-α), resulting in damage to the epithelial and endothelial cells, fibroblasts, and extracellular matrix components [66].

Cell proliferation is due to the inhibition of apoptosis by multiple mechanisms regulated by different microorganisms [72].

The production of carcinogens is recognized in the transformation of alcohol to acetaldehyde, production of organic acids, volatile sulfuric compounds, and ROS [73].

According to studies based on genomics, most human cancer contains a specific intratumoral MB, which varies between different subjects for species, role in the development of cancer itself, and the treatment response [74,75,76]. The intratumoral MB is well organized in micro niches with immune and epithelial cell activities that favour cancer growth rather than being distributed randomly within patient tumours [51]. Analyses performed at the tumour microenvironment, in the case of OSCC, revealed *Parvimonas*, *P. gingivalis*, and *Fusobacterium* as dominant genera. Analyses performed by a transmission electron microscope, in the case of OSCC, revealed *Parvimonas*, *P. gingivalis,* and *Fusobacterium* as dominant genera [51]. Analyses performed on single cells also showed intercellular variability related to bacterial load, surrounding inflammatory pattern, and genera. These bacteria can alter the transcriptional program of some cells causing cellular heterogeneity in cancer cells [51].

Probiotic administration could be considered to treat the consequences of thyroid hormone withdrawal (THW). Lin et al. intended to assess the characteristics of oro-intestinal microbiota in THW patients. Thyroid carcinoma is the most frequent endocrine system cancer. Radioactive iodine absorption in patients depends on thyrotropin stimulation, which can be accomplished by THW [52]. Nevertheless, THW is usually associated with several problems, such as tiredness, constipation, weight gain, edema, and hypercholesterolemia, all of which significantly influence the quality of life of patients [77,78,79]. This study aimed to see if probiotics may help with THW problems and if these therapeutic benefits were connected to the state of the oro-intestinal flora. The probiotic mixture used included *Bifidobacterium infantis*, *Lactobacillus acidophilus*, and *Lactobacillus casei* [52]. The research indicates that probiotic treatment raised the diversity of the gut MB while decreasing the number of inflammatory bacteria such as *Fusobacterium* and enhancing the energy metabolism pathway [52]. Additionally, probiotic therapy decreased fecal and plasma LPS levels, indicating that probiotics may enhance intestinal barrier function. As a result, there was a decrease in the frequency of fatigue, weight gain, constipation, and dry mouth, but there was no modification in the edema [52].

Another possible application of probiotics for preventive purposes is their use in trying to prevent radio-chemotherapy (RCHT)-induced OM in HNC. The incidence of OM was examined in research carried out by De Sanctis et al. comparing patients treated with *Lactobacillus brevis CD2* (LB CD2) lozenges and those treated with sodium bicarbonate mouthwash after radical concomitant RCHT for HNC [45]. Taking into consideration the major limitation of this study of not reaching the established number of patients due to a lack of LB CD2 supplies, no benefit was observed in using LB CD2 in reducing the incidence of oral mucositis [45]. However, the idea that diseases such as OM can be linked to a change in the microbial makeup of the biofilm that colonizes the oral cavity is generating more and more interest. RCHT or the tumour itself may affect the salivary bacterial ecology, causing mucosal injury and modifying the numbers of opportunistic microorganisms that may become pathogenic in cancer patients who develop OM [45].

### 4.3. Microbiota and Cancer Treatment

As the central role of the MB in the regulation of physiological and even pathological functions becomes increasingly clear, it is expected to assume increasing significance in the prevention and treatment of various conditions, including cancer [80]. 

There is nowadays evidence to support the concept that the MB can affect how patients respond to cancer treatment. Several studies have demonstrated that the MB might modify the effectiveness of cancer therapy, including chemotherapy and immunotherapy, and susceptibilities to collateral effects [50,81,82,83,84]. 

Commensal bacteria have recently been demonstrated to enhance the effects of immunotherapy with checkpoint inhibitors. Dietary supplementation with *Bifidobacterium* increases tumour control to the same extent as programmed death-ligand 1 (PD-L1)-specific antibody therapy, and combination therapy almost eliminated tumour expansion [85]. Similar to this, different *Bacteroides* species have different impacts on the immunostimulatory effects of Cytotoxic T-Lymphocyte Associated Protein 4 (CTLA-4) inhibition, which result in anticancer consequences [86].

The combination of immunotherapy and APG-157, derived from the plant curcuma longa, proved to be an effective therapeutic strategy. In fact, following the administration of curcumin, in the two groups that received two different dosages in the form of tablets (100 mg and 200 mg), the concentration of pro-inflammatory blood and salivary cytokines was significantly reduced, compared with the group that received placebo. This positively changes the microbiota and reduces tumour cell senescence [49].

As it is known, the MB in cancer patients is different from that in healthy subjects [87].

The intratumoral MB also changed significantly after APG-157 administration, as there was evidence of a reduction in pathogenic *Bacteroides* species [49].

Previous studies in mice and human trials have shown a direct association between *Bacteroides* and an increased risk of colon cancer development. Thus, APG-157 could have an antitumor effect by modifying the MB and attracting immune response cells [88].

Preliminary studies imply that the regulation of the MB may develop into a cutting-edge method for enhancing the effectiveness of immune-based cancer therapy [89].

Moreover, the use of probiotics as adjuvant therapy in cancer patients also showed excellent results [80].

Probiotics may, because of their qualities, lower the toxicity of RCHT, according to experimental data and some clinical evidence. This, in turn, strengthens homeostasis and lessens the side effects of cancer treatment [76,81].

In the treatment with radiation for HNC, it is impossible to avoid the exposure of major salivary glands; thus, the production of saliva changes [90]. Hyposalivation inevitably alters the microflora of saliva: large salivary Glycoproteins (such as immunoglobulin A), coating the oral mucosa, act as a barrier for the surface cells and decrease the adhesion of bacteria to the oral mucosa. When this protective property of the glycoproteins is absent, due to lack of salivation, the epithelial cells are more prone to irritation and oral problems, such as dental cavities, mucositis, and candidiasis [86,86,91,92]. 

The microbial chances may be directly correlated to ionizing radiation exposure [93,94]. 

An analysis of saliva samples collected from eight subjects between 26 and 70 years before and during radiation therapy, following amplification of the V1-V3 hypervariable regions of bacterial 16S rRNA genes, revealed fluctuations in the composition of the supragingival flora [44].

While some phyla and genera varied at different stages of therapy, 11 (Streptococcus, Actinomyces, Veillonella, Capnocytophaga, Derxia, Neisseria, Rothia, Prevotella, Granulicatella, Luteococcus, and Gemella) out of 140 genera and 4 (Actinobacteria, Bacteroidetes, Firmicutes, and Proteobacteria) out of 13 phyla, were found in all subjects in all stages, suggesting the concept of a core microbiome and the future possibility to predict community responses to disturbances caused by exposure to ionizing radiation [44].

Additionally, it was noted that the number of operational taxonomic units changed throughout radiotherapy. As the radiation dose was raised, there may be a general decline in the number of species, according to the negative association between the number of units and dosage [44].

One of the most common complications after radiotherapy is OM, which generally develops two or more weeks after the treatment [95]. 

In patients receiving concurrent RCHT or radiotherapy with changed fractionation, the average incidence of OM is about 80% [96].

Mucositis severity is correlated with poor oral hygiene, oral candidosis, and xerostomia [97]. 

Dietary supplementation of a cocktail of probiotics (based on *L. plantarum*, *B. animalis*, *Lactis*, *L. rhamnosus* and *L. acidophilus*) revealed an overall reduction in the incidence of OM in patients undergoing combined treatment (chemo-radio-probiotic), as well as possible side effects of the treatment itself, related to an improvement in the individual’s immune response [50].

Consistent results were shown in Jiang’s study, in which only 15.52% of patients with nasopharyngeal carcinoma who received combined RCHT and probiotics (*Bifidobacterium longum*, *Lactobacillus lactis*, and *Enterococcus faecium*) developed grade 3 or 4 OM, compared with 45.71% of patients who received the placebo together with radio- and chemotherapy treatment instead. Additionally, in this study, as in the previous one, patients were monitored from the beginning up to 7 weeks after the end of the therapy [46].

*Lactobacillus brevis*, on the other hand, alone, can reduce advanced-grade OM in treated patients suffering from HNSCC [98].

The study also evidenced that decreased levels of leukocytes (23.5% vs. 23.0%), haemoglobin (23.2% vs. 18.4%), erythrocytes (12.3% vs. 10.3%), and lymphocytes (30.6% vs. 30.8%) were the most common side effects (8%) caused by the treatment [46].

Probiotic administration improved CD4+, CD8+, CD3+, the haemoglobin and lymphocytes ratio, and influences the production of immunoglobulin; however, it had no significant effect on weight loss or bone narrow suppression [46,99]. Enhancement of the immune system by administrating probiotics allows for a reduction in the occurrence of oral mucositis. This is also possible since probiotics work to restore the gut MB and were found to significantly decrease the toxicity of radiotherapy and chemotherapy, increasing food digestion, patients’ energy and immunity; specifically, *B. longum*, *L. lactis*, and *E. faecium* should be given more consideration in the future for the purposes of a therapeutic strategy [46].

On the other hand, the MB also changes with oral carcinoma in patients who underwent surgical treatment [52]. Although there was no significant difference in the total bacterial density of saliva, the bacterial composition was found to be significantly different after the surgery. Some bacteria arising from dental plaque, including periodontal pathogens, increased significantly, suggesting the need for more careful and frequent postoperative oral care [47].

Patients with unresectable oesophageal cancer receiving chemotherapy and radiation therapy frequently experience radiation esophagitis (RE), a common treatment-emergent adverse event that causes significant morbidity and mortality. In the current investigation, we sought to determine if oral bacterial diversity and RE were correlated in chemoradiotherapy-treated oesophageal cancer patients [48].

With recent advances in next-generation sequencing with 16S rRNA gene amplicon analysis, the correlation between the OMB and systemic diseases, including diabetes, inflammatory bowel disease, obesity, and cancer, has been widely recognized and characterized. Therefore, a prospective study using high-throughput sequencing of 16S rRNA gene amplicon analysis confirmed the association between RE and the OMB [48]. 

The microbiome exhibits high inter-individual variation as well as macro and microscopic structural variability, which is influenced by variables including age, gender, BMI, food, and antibiotic usage. When comparing healthy controls to colorectal patients, these parameters should preferably be added into the studies using group matching, but the majority of the selected publications simply provided potential influencing variables on microbiome composition without actual comparison of the two groups. Pilot studies have also discovered considerable differences amongst tumors, in part because the gut microbiome is dynamic and develops with disease, and in part because it is prone to confounding environmental influences including nutrition, medicine, smoking, and other lifestyle factors. These were seldom taken into consideration in the analysis, and most were not published. The richness of the faecal microbiome differs at the phylum and strain levels between geographically distinct populations and between nations. Because the bulk of the included publications were conducted in Asia, translation to other locations may be limited, and it is uncertain how many of these studies are translatable. Another significant problem in these investigations has been the lack of consistency in sample collection and processing. Sample collection varied greatly between investigations (for example, before or after bowel-cleansing agents necessary for the colonoscopy). These variables are known to have an impact on microbiome makeup. Furthermore, DNA extraction was performed using a variety of DNA extraction kits, adding another variable impacting microbiome makeup. The analytical methodologies used by the included research also lacked uniformity. To identify bacteria, each study utilized a different analytical approach, either 16S rRNA or metagenomics. The majority of 16S rRNA analysis is dedicated to taxonomic analysis with an assumed functional interpretation. Although metagenomics allows for species-level identification and profound functional insights, it is not currently inexpensive for population-level investigations, and its interpretation takes substantial computational capacity. There was significant variation in the primers employed within these approaches, resulting in potential biases and making comparisons between them difficult.

## 5. Conclusions

The modern approach of precision medicine based on the objective study of DNA, RNA, epigenetics, proteins, and metabolites ensured a reliable analysis of the MB with positive implications for patients’ systemic health. 

Alterations in OMB are appreciated also in patients with head and neck cancer such as an increase in potential pathogens and a decrease in bacteria physiologically present in the oral cavity. Differences in oral microbiota can influence also the outcome of patients with oral squamous cell carcinoma because *Fusobacterium* is associated mostly with a better outcome. Confirming the correlation between microbiota and patient’s health, the investigation of gut microbiota composition can also be useful for patients with head and neck cancer. 

The possibility of creating a database that will lead clinics to diagnosing and treating oral diseases and cancers through precision medicine gives, as well as economic advantages to the health care system, a predictable and rapid management of the patient.

## Figures and Tables

**Figure 1 jpm-13-00933-f001:**
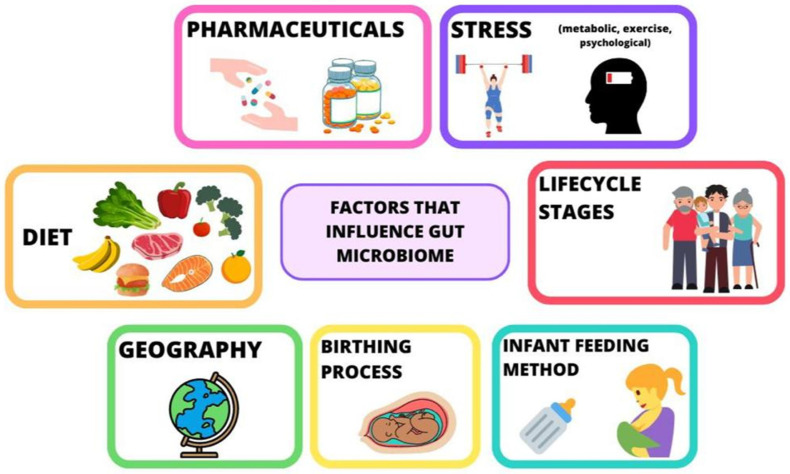
Factors influencing gut microbiome.

**Figure 2 jpm-13-00933-f002:**
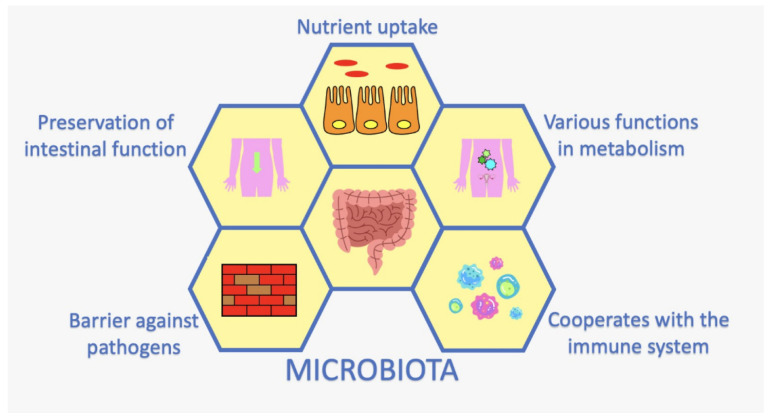
Microbiota pluripotency on human metabolism.

**Figure 3 jpm-13-00933-f003:**
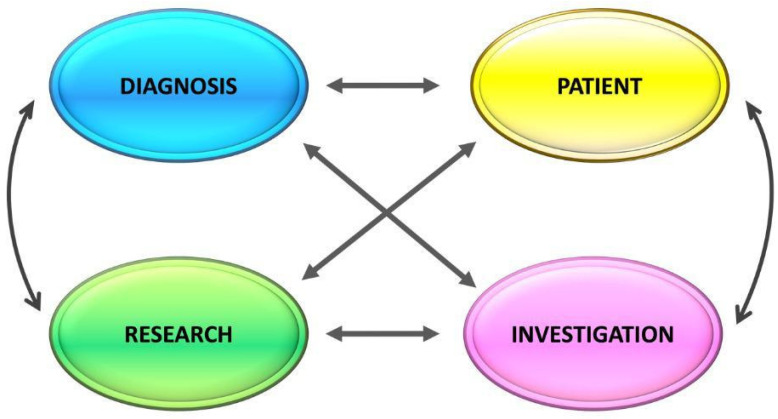
New biomedicine approaches loops.

**Figure 4 jpm-13-00933-f004:**
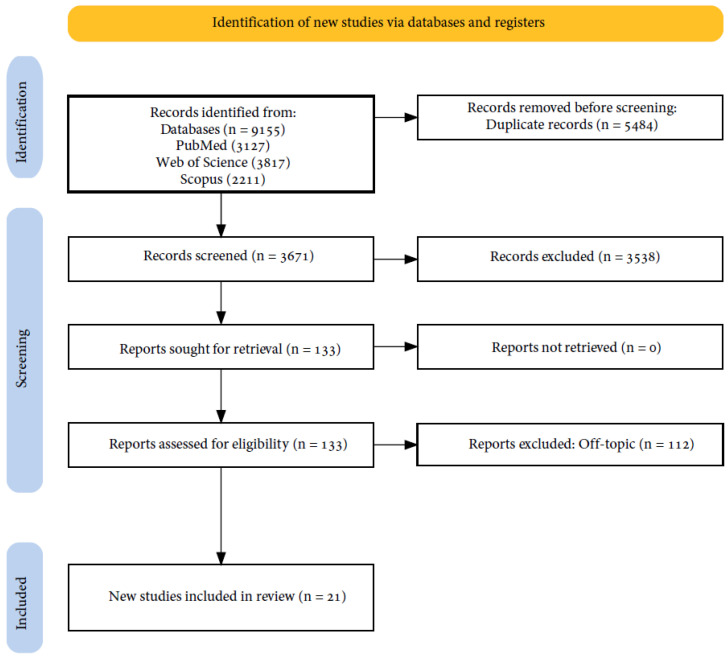
PRISMA flowchart diagram of the inclusion process.

**Table 1 jpm-13-00933-t001:** PICOS criteria.

Criteria	Application in the Present Study
**Population**	Subjects diagnosed with head–neck tumors and potentially malignant lesions
**Intervention**	Use of cancer biomarkers. Supplementation with probiotics or prebiotics or symbiotic
**Comparisons**	Comparing before and post cancer intervention MB, MM assessment determined by 16 s rRNA sequencing, Mann–Whitney test on cancer biomarkers: fluorodeoxyglucose-positron emission tomography (FDG-PET/CT) scans
**Outcomes**	Changes in baseline and end in symptom measurements. Changes in MB composition due to the presence of cancer or after cancer treatments
**Study design**	Clinical Trials

**Table 2 jpm-13-00933-t002:** Characteristics of the included studies.

Authors	Study Type	Aim of the Study	Materials and Methods	Results
Swartz et al., 2022 [37]	Case-control Study	Tumor hypoxia compromises local control and patient survival. We used a digital, single-cell analysis to compare two hypoxia biomarkers (hypoxia-inducible factor 1-alpha [HIF-1] and pimonidazole [PIMO]) and their effect on outcome in patients with laryngeal cancer who received accelerated radiotherapy with or without carbogen breathing and nicotinamide (AR versus ARCON).	HIF-1 and PIMO immunohistochemical labeling were carried out in successive sections of 44 laryngeal carcinoma patients randomized between AR and ARCON. HIF-1 expression and PIMO-binding were associated in QuPath utilizing digital image analysis. Each biomarker’s high-density regions were automatically identified, and staining overlap was examined. For each biomarker, Kaplan–Meier survival analyses for local control, regional control, and disease-free survival were performed to predict a response advantage of ARCON over AR alone.	In total, 106 tissue pieces from 44 individuals were examined. On the fragment level, a weak but significant positive connection was found between HIF-1 and PIMO positivity, but not at the patient level. The number of high-density staining patches for both biomarkers showed a moderately strong connection (r = 0.705, p = 0.001). The staining overlap was inadequate. ARCON’s response benefit over AR could not be predicted by HIF-1 expression, PIMO-binding, or a combination of the two.
Nagao et al., 2017 [38]	Randomized Controlled Trial	The goal of this study was to see if there were any variations in baseline p53 and ki67 expression between those who responded and those who did not react to our intervention. A secondary goal was to determine whether there was a link between dietary parameters and clinical responses.	We included all nonsmokers in the experimental group (n = 23) for this biomarker investigation. At the 1-year follow-up, there were four responders and 12 non-responders among the 16 who completed the experiment for one year of supplementation. Following p53 and ki67 immunostaining, the proportion of positive cell nuclei was calculated as the labeling index (LI).	The expression of p53 was higher in the basal layers than in the para-basal layers. Non-responsive subjects had a greater mean para-basal LI of p53 (26.0) than responding subjects (11.2) (p = 0.028). The ki67 LIs in the two groups were not substantially different.
Ganly et al., 2019 [39]	Case-control Study	The goal of this study was to see if the oral was linked to OC-SCC in nonsmokers with HPV negative. We investigated the oral MMs of HPV-negative nonsmokers with OC-SCC (n = 18), premalignant lesions (n = 8), and healthy controls (n = 12).	Their oral MB was obtained using an oral wash and characterized using 16S rRNA gene sequencing.	In OC-SCC, the periodontal pathogens *Fusobacterium*, *Prevotella*, and *Alloprevotella* were abundant, but commensal *Streptococcus* was decreased. We divided the oral MM into two categories based on the four species plus a marker genus *Veillonella* for PML.
Ganly et al., 2021 [40]	Case-control Study	The goal of this study was to use metagenomic shotgun sequencing to compare the HPV genome in patients with oral cavity squamous cell carcinoma (OCSCC) to normal people.	They gathered 50 OCSCC patients and compared them with a control patient based on age, gender, race, smoking status, and alcohol status. All patients’ DNA was collected from oral wash samples and full genome shotgun sequencing was carried out. The raw sequencing data was cleaned, reads were matched with the human genome (GRCH38), nonhuman reads were detected, and HPV genotypes were determined using HPViewer. The tongue was the most prevalent subsite in 26 (52% of the 50 individuals with OCSCC). Primary resection and neck dissection were performed in all patients.	p16 immunohistochemistry was negative in all but two tumors. In terms of gender, age, race/ethnicity, alcohol use, and cigarette smoking, there were no statistically significant differences between the cases and controls. In the nonhuman DNA readings, there was no statistically significant difference between the cancer samples and the control samples. HPV was found in 5 instances (10%) of OCSCC (genotypes 10, 16, and 98), although only 1 tumor sample (genotype 16) produced enough reads to imply an involvement in the genesis of OCSCC. HPV was found in four healthy people, however each had just 1–2 HPV readings per human genome. HPV genotypes are uncommon in patients with oral cancer.
Torralba et al., 2021 [41]	Prospective Observational Study	To elucidate the links between the oral MB and cancer virulence factors	They employed 16S rDNA and metagenomic sequencing to evaluate the microbial makeup and functional content of 18 OSCC patients’ tumor tissue, non-tumor tissue, and saliva.	When compared with all other sample categories, the results show a larger number of bacteria from the phyla *Fusobacteria*, *Bacteroidetes*, and *Firmicutes* linked with tumor tissue. Furthermore, saliva metaproteomics indicated a substantial rise in *Prevotella* in five OSCC patients, whereas *Corynebacterium* was predominantly related to 10 healthy patients. Finally, we discovered adhesion and virulence factors linked with *Streptococcus gordonii* as well as recognized oral pathogens belonging to the *Fusobacterium* genera, which were primarily detected in OSCC tissues.
Garajei et al., 2023 [42]	Case-control Study	This study compares the expression of the miR-21-5p and miR-429 genes in biopsy samples from patients with OSCC to that of controls.	Tissue samples were collected from 40 people (20 OSCC patients and 20 healthy controls) and analyzed using the Mann–Whitney test to evaluate miR-21-5p and miR-429 expression.	The individuals in the control and sick groups were 47.15 and 53.8 years old, respectively. The Mann–Whitney test revealed significant differences in miR-21-5p (p = 0.0001) and miR-429 (p = 0.0191) expression levels between the two groups (p = 0.05).
Shah et al., 2022 [43]	A retrospective study of serial FDG-PET/CT scans collected prospectively as part of a phase 2 open-label randomized clinical trial examining neoadjuvant immunotherapy in patients with untreated OCSCC between 2016 and 2019 was performed.	To connect variations in fluoro-[18F]FDG-PET/CT scans with primary tumor pathologic response and immunologic biomarkers in patients with OCSCC undergoing neoadjuvant immunotherapy.	In total, 29 patients with untreated OCSCC (T2, or clinically node positive) from a single academic medical center were randomized 1:1 to receive neoadjuvant therapy with single agent nivolumab or combination nivolumab and ipilimumab, followed by surgery and standard of care adjuvant therapy. FDG-PET/CT scans were performed before (T0) and after (T1) preoperative immunotherapy in this investigation.	There was no relationship between pathologic response and SUVmax change in primary OCSCC between T0 and T1. Thirteen of the 27 subjects had newly FDG-avid ipsilateral LNs at T1, with the majority being pathologically negative. A total of 9 patients experienced radiologic irAEs, the most frequent of which was sarcoid-like LN (7 of 27). There were no relationships between primary OCSCC SUVmax at T0 and CD8+ T-cell number in the main tumor biopsy, and there were no associations between primary OCSCC SUVmax at T1 and CD8+ T-cell number in the original tumor during surgery.
Hu et al., 2013 [44]	Prospective study	The purpose is to investigate the dynamic core microbiome of oral microbiota in supragingival plaque during head-and-neck radiation.	Dental plaque samples were collected from 8 subjects before and during radiotherapy.	During radiation, 4 phyla and 11 genera were detected, validating the hypothesis of a core microbiome.
De Sanctis et al., 2019 [45]	Clinical Trial	To evaluate the effect of lactobacillus brevis CD2 (LB CD2) in preventing oral mucositis in patients with head and neck cancers (HNC).	In total, 75 patients were included to receive either LB CD2 lozenges or a mouthwash routine with sodium bicarbonate.	The trial failed to establish the effectiveness of LB CD2 in reducing radiotherapy-induced OM.
Jiang et al., 2019 [46]	Randomized Controlled Trial	Probiotics will be used to decrease the severity of OM caused by chemoradiotherapy in patients with nasopharyngeal cancer.	During radiotherapy, 99 patients were randomly randomized to receive a probiotic or a placebo.	The gravity of OM was significantly reduced in those who took the probiotic combination.
Kageyama et al., 2020 [47]	Prospective study	To evaluate the compositional shift of oral microbiota after surgical resection of tongue cancer.	Saliva samples were collected from 25 tongue cancer patients before and after resection of the tongue. Quantitative PCR analysis and 16S ribosomal RNA (rRNA) gene sequencing were used to determine bacterial density and composition.	The surgical resection of the tongue caused a shift in the structure of the salivary microbiota, with an increase in bacterial species from dental plaque, especially periodontal pathogens.
Xu et al., 2020 [48]	Pilot study	The objective is to examine the relationship between oral bacterial variety and radiation esophagitis in chemoradiotherapy patients with esophageal cancer.	Oral mucosal swabs were obtained from 10 patients who did not have RE, 11 patients who had grade 1 RE, and 10 patients who had grade 2 RE. The diversity of oral bacteria was measured using 16S rRNA gene sequencing.	In patients with esophageal cancer following chemoradiotherapy, a reduction in oral bacterial diversity may be associated with RE.
Basak et al., 2020 [49]	Randomized Controlled Trial	The purpose of this research is to evaluate the effect of APG-157 on cytokines and microbiota.	APG-157 was given to 13 healthy people and 12 patients with oral cancer. Blood and saliva samples were collected before, 1, 2, 3, and 24 h after therapy.	This study shows that APG-157 is a strategic therapy combined with immunotherapy in cancer.
Xia et al., 2021 [50]	Randomized Clinical Trial	The purpose is to use Probiotics to prevent radiochemotherapy-induced OM in patients with nasopharyngeal carcinoma.	In total, 77 individuals were chosen and randomly selected to receive either a probiotic cocktail or a placebo. After 7, 14, and 21 days, tongue, blood, fecal, and proximal colon tissue samples were examined.	The improved probiotic cocktail considerably decreases the severity of OM by improving patients’ immune responses and changing the composition of their gut microbiota.
Niño et al., 2022 [51]	Observational study	To evaluate the effect of the intratumoral microbiome on cancer spatial and cellular heterogeneity.	Spatial profiling and single-cell RNA sequencing are used in situ to identify cellular, molecular, and spacial host-microbe interactions.	Inside a tumor, the microbiota is well structured in microniches with immune and epithelial cell activities that support cancer growth.
Lin et al., 2022 [52]	Randomized Clinical Trial	To assess the oral-gut microbiota profiles of THW patients and then see if probiotics can help with THW-related problems.	In total, 50 thyroid cancer patients were randomly randomized to receive probiotics or a placebo during thyroidectomy.	Probiotics significantly improved gut and oral microbial diversity and reduced thyroid hormone withdrawal-related problems in thyroid cancer patients prior to radioiodine treatment after thyroidectomy.
Li et al., 2023 [53]	Observational study	Based on tissue sequencing, this study intends to assess the clinical association of the intratumoral oral microbiome in oral squamous cell cancer.	The oral microbiota was analyzed in 133 OSCC samples to assess its composition compared with healthy patients and also determine its diagnostic and prognostic value.	Differences in bacterial composition have been found between the oral microbiota in OSCC and healthy patients, and only some of these bacteria can be used as diagnostic and prognostic predictors.
Li, Z. et al., 2021 [54]	Prospective study	To assess the prevalence and distribution of the oral microbiota in oral cancer patients, populations without precancerous lesions, and healthy individuals. To assess the connection between oral cancer incidence and the microbiome of oral bacteria.	In total, 10 patients who had been diagnosed with oral cancer, 10 healthy subjects and 6 patients with oral precancerous lesions were enrolled. Salivary samples were collected from these patients and the microbiome was analyzed	At many species levels, there were significant structural alterations in the oral microbiota of patients with oral cancer, patients with precancerous lesions, and healthy controls. Some metabolic pathways are altered by dysbiosis of the oral microbiota, which has an impact on oral health.
Chan, J.Y.K. et al., 2022 [55]	Prospective cohort study	The purpose of the study is to understand the correlation between oral microbiome, HPV infection, conventional risk factors, and head and neck squamous cell cancer (HNSCC).	By sequencing 16S rRNA V3-V4 bacterial and HPV L1 sections, respectively, the oral microbiota and HPV infection of tissues of 166 Chinese people were analyzed. the relationship between oral microbiota, HPV and clinical features was analyzed.	It is inferred that dysbiosis of the oral microbiota is involved in the pathogenesis of OSCC. *Fusobacterium* is involved in improving outcomes of patients with OSCC, especially in patients without traditional risk factors. Understanding how the oral microbiome, HPV infection, and other risk factors for HNSCC interact will be crucial to understanding the pathophysiology of this disease.
Zuo, H.-J. et al., 2020 [56]	Prospective study	In this study, oral microbial traits and new biomarkers will be assessed for HNC patients, and the relationship between oral microorganisms and HNC-related symptoms will be evaluated prior to surgical intervention.	Overall, 56 patients with HNC and 64 healthy controls were recruited. Salivary samples were taken in order to do 16S rRNA gene sequencing on the microbes.	A decrease in health-related bacteria, such as *Peptococcus*, and a rise in potentially pathogenic bacteria, such as *Capnocytophaga* and other LPS-producing bacteria such as *Neisseria*, were seen in the oral microbiome of HNC patients. Additionally, HNC-related symptoms in conjunction with salivary microorganisms such as *Capnocytophaga* may be employed as a noninvasive technique for screening, identification, and treatment monitoring of HNC.
Al-Qadami, G. et al., 2023 [57]	Prospective pilot study	To assess the relationships between a patient’s pre-treatment gut flora and the severity of radiotherapy-induced oral mucositis (OM) and recurrence risk in those with head and neck cancer (HNC).	Patients who were scheduled to receive radiotherapy or chemoradiotherapy for HNC were enrolled in this trial. Before therapy, stool samples were taken, and 16S rRNA gene sequencing was used to analyze the microbial composition.	OM severity and recurrence risk are related to a patient’s gut microbiota makeup at the beginning of therapy.

## Data Availability

Not applicable.

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
