# Peer review of "Precision Medicine on the Effects of Microbiota on Head–Neck Diseases and Biomarkers Diagnosis"

_jpm, 2023, doi:10.3390/jpm13060933_

Round 1

Reviewer 1 Report

The manuscript entitled “Precision Medicine on the Effects of Microbiota on Head-Neck  Diseases and Biomarkers Diagnosis. A Systematic Review” discusses a relevant topic. However, the authors should consider some important points to improve the manuscript. It is also necessary to review the English language because some inaccuracies were observed in the text.

 1.       Abstract:

We suggest to improve the description of the Results. A brief paragraph about the profile of the manuscripts that had been selected must be added to the text. As a matter of fact, it seems to me that some relevant data are described in Conclusions. Please, re-write it.

2.       Introduction:

Have the figures been created by the authors themselves? It is necessary to add the appropriate source;

“In the presence of dysbiosis, inflammatory and metabolic diseases (obesity, hyper-tension, ulcerative colitis, diabetes mellitus, heart disease, and oncological diseases) as well as inhibition or reduction of 'absorption of drugs thus affecting the course of the disease”. The idea of this paragraph is not clear. Please, clarify the “course of the disease”. What disease are the authors specifically mentioning?

The Introduction is too long. Some paragraphs are too short and we suggest to link them.

Perhaps, it would be better to discuss the occurrence of cancer by the action of the oral MB in Discussion.

3.       Material and Methods

The authors mentioned that two separate re-viewers (F.P. and A.P.) evaluated the retrieved studies for inclusion using the criteria, and disagreements were addressed by consensus. We suggest applying some index in order to check the level of agreement.

We suggest to use a scale to analyze the clinical trials quality. https://doi.org/10.1016/0197-2456(95)00134-4.

4.       Results

 The table format is not appropriate.

 5.       Discussion

The Discussion is too long. The authors have to highlight the mean points and discuss about the limitations of their systematic review.

 It will be necessary to review the English language because some inaccuracies were observed in the text.

Author Response

Please, view the attached file

Reviewer 2 Report

In this manuscript, the author wants to put all the information in this review. However, it may render the reader out off the focus. It would be better if the author could focus on one certain disease like oral cancer and survey the dysbiosis related articles to this disease. Besides, there is no any related data about the analysis of the reviewed articles. The author should show some results mentioned in the manuscript.

Author Response

Please, view the attached file

Round 2

Reviewer 1 Report

I noticed that the authors have made all the corrections. The manuscript is suitable now.